# Facial Recognition Intensity in Disease Diagnosis Using Automatic Facial Recognition

**DOI:** 10.3390/jpm11111172

**Published:** 2021-11-10

**Authors:** Danning Wu, Shi Chen, Yuelun Zhang, Huabing Zhang, Qing Wang, Jianqiang Li, Yibo Fu, Shirui Wang, Hongbo Yang, Hanze Du, Huijuan Zhu, Hui Pan, Zhen Shen

**Affiliations:** 1Eight-Year Program of Clinical Medicine, Peking Union Medical College Hospital, Peking Union Medical College, Chinese Academy of Medical Sciences, Beijing 100730, China; danie_wu@student.pumc.edu.cn (D.W.); pumcfyb@163.com (Y.F.); 2Department of Endocrinology, Key Laboratory of Endocrinology of National Health Commission, Translation Medicine Centre, Peking Union Medical College Hospital, Peking Union Medical College, Chinese Academy of Medical Sciences, Beijing 100730, China; cs0083@126.com (S.C.); huabingzhangchn@163.com (H.Z.); wangsr13@126.com (S.W.); yanghb@pumch.cn (H.Y.); vespasian_du@126.com (H.D.); shengxin2004@163.com (H.Z.); 3Medical Research Center, Peking Union Medical College Hospital, Peking Union Medical College, Chinese Academy of Medical Sciences, Beijing 100730, China; yuelunzhang@outlook.com; 4Department of Automation, Tsinghua University, Beijing 100084, China; qing.wang@tsinghua.edu.cn; 5School of Software Engineering, Beijing University of Technology, Beijing 100124, China; lijianqiang@bjut.edu.cn; 6Key Laboratory of Endocrinology of National Health Commission, Department of Endocrinology, State Key Laboratory of Complex Severe and Rare Diseases Peking Union Medical College Hospital, Chinese Academy of Medical Sciences and Peking Union Medical College, Beijing 100730, China; 7State Key Laboratory for Management and Control of Complex Systems, Beijing Engineering Research Center of Intelligent Systems and Technology, Institute of Automation, Chinese Academy of Sciences, Beijing 100190, China; 8Qingdao Academy of Intelligent Industries, Qingdao 266109, China

**Keywords:** artificial intelligence, computer-aided diagnosis, facial phenotypes, machine learning, complexity theory

## Abstract

Artificial intelligence (AI) technology is widely applied in different medical fields, including the diagnosis of various diseases on the basis of facial phenotypes, but there is no evaluation or quantitative synthesis regarding the performance of artificial intelligence. Here, for the first time, we summarized and quantitatively analyzed studies on the diagnosis of heterogeneous diseases on the basis on facial features. In pooled data from 20 systematically identified studies involving 7 single diseases and 12,557 subjects, quantitative random-effects models revealed a pooled sensitivity of 89% (95% CI 82% to 93%) and a pooled specificity of 92% (95% CI 87% to 95%). A new index, the facial recognition intensity (FRI), was established to describe the complexity of the association of diseases with facial phenotypes. Meta-regression revealed the important contribution of FRI to heterogeneous diagnostic accuracy (*p* = 0.021), and a similar result was found in subgroup analyses (*p* = 0.003). An appropriate increase in the training size and the use of deep learning models helped to improve the diagnostic accuracy for diseases with low FRI, although no statistically significant association was found between accuracy and photographic resolution, training size, AI architecture, and number of diseases. In addition, a novel hypothesis is proposed for universal rules in AI performance, providing a new idea that could be explored in other AI applications.

## 1. Introduction

Many diseases display distinctive facial manifestations, especially endocrine diseases and genetic diseases, including monogenic disorders, chromosomal diseases, and thousands of rare diseases [1]. Recognition by the human eye often causes misjudgment and delays diagnosis due to inconspicuous early facial symptoms associated with these diseases, large individual facial differences, and lack of physicians’ knowledge of rare diseases. With the development of artificial intelligence (AI) technology, AI methods have been widely applied in different fields [2,3,4,5,6]. Automatic image recognition based on AI could identify image features for the diagnosis and screening of various diseases, with satisfactory performance for the diagnosis of pulmonary nodules, tumors, fundus diseases, even COVID-19 [7,8,9,10]. Among these AI techniques, facial recognition based on artificial intelligence enables computers to detect underlying facial patterns and has played an important role in the diagnosis and screening of diseases with facial phenotypes or changes in recent years [11,12]. It is assumed that artificial intelligence could help to improve diagnostic accuracy and to avoid delayed diagnosis, leading to earlier intervention, conservation of social healthcare resources, and implementation of health policies in the future [12,13,14]. Different models and systems have been developed to provide possible improvement for diagnostic accuracy [15].

However, there remains a lack of exploration of the factors influencing AI performance or of universal rules to reduce heterogeneity [14]. As has been shown before, diagnostic accuracy of facial recognition for Turner syndrome tended to be lower than that of Down syndrome, although a larger sample size helped to improve it [16,17]. However, the heterogeneity of diseases and AI methods studied and the limited number of works on rare diseases makes it difficult to review and summarize individual studies in a unified manner. Since the complexity theory could be applied to quantitatively describe facial features, this theory needs to be developed to explore the universal rules determining the diagnostic performance of AI based on facial features for heterogeneous diseases.

This is the first study that conducted a systematic review and meta-analysis to summarize the data regarding the diagnosis of heterogeneous diseases on the basis of facial features and explored the universal rules governing the application of facial recognition based on AI in the field of medical diagnosis. We aimed to quantitatively analyze the diagnostic accuracy of facial recognition based on AI, as well as the factors influencing the diagnostic performance and to provide a potential reference for clinical practice. In addition, our study proposes a potential hypothesis for evaluating the performance of AI in other fields, such as image recognition based on AI, and provides a new idea for dealing with heterogeneity when reviewing and analyzing the performance of AI applications.

## 2. Materials and Methods

### 2.1. Study Identification and Selection

We searched Medline, PubMed, IEEE, Cochrane Library, EMBASE to identify potential eligible studies published from 1 January 2010 to 15 August 2021. The references of relevant publications were also checked manually. The detailed search strategy containing the index test (facial recognition) and the target condition (diagnosis) is shown in Appendix A.

Studies were included if they evaluated facial recognition by algorithms of artificial intelligence for the diagnosis of diseases based on facial phenotypes or deformities using photographs and provided sufficient information for quantitative data synthesis. Studies were excluded of they were reviews, lacked a control group, or identified more than one possible disease as a diagnostic result by facial recognition. The titles and the abstracts were screened by two reviewers independently (DW and SC), and the full texts of potentially eligible studies were further screened.

### 2.2. Data Extraction and Quality Assessment

The data obtained from each study included publication characteristics (authors and year of publication); characteristics of the targeted disease (number of diseases and specific facial features); characteristics of the sample set (data sources, age, sex, and resolution of photographs); characteristics of the index test (algorithms, and number of images used in model training); characteristics of the reference standard (diagnostic criteria); accuracy data (number of true positives, true negatives, false positives, and false negatives). Supplements in each study were also reviewed if available.

Quality Assessment of Diagnostic Accuracy Studies-2 (QUADAS-2) was used to assess the risk of bias in patient selection, index test, reference standard, and flow and timing of the included studies. Publication bias was not assessed in our study because there is not a universally accepted method for the review of diagnostic studies to detect publication bias according to the Cochrane Handbook for Diagnostic Tests Review.

### 2.3. Definition and Calculation of FRI

We defined facial recognition intensity (FRI) as an index to describe the difference of facial features between a studied disease and healthy controls. FRI is calculated as shown in Equation (1) by multiplying the number of independent facial phenotypes of a disease and the maximum penetrance among these facial features.
FRI = Nf × Pmax(1)

In Equation (1), Nf represents the number of facial phenotypes relevant to a disease, and Pmax is the maximum penetrance among these facial features, representing the percentage of individuals in a group of patients who exhibited a specific facial phenotype. The facial features and the penetrance of facial phenotypes were collected from the original articles and relevant reviews. If a facial phenotype was associated with a specific group of patients, penetrance was defined to be 100%. Since some of the facial phenotypes were correlated, such as small jaws and crowded teeth, associated phenotypes were counted only once to calculate FRI. For example, Down syndrome displayed nine independent facial phenotypes, and the maximum penetrance of these facial phenotypes was 100% [18]; hence, FRI of Down syndrome was calculated by multiplying 9 by 100%, resulting in 9. FRI was defined to summarize the common characteristics of objects, e.g., facial phenotypes in the presence of different diseases, and to minimize heterogeneity among objects analyzed by AI methods so to make them comparable in the subsequent analysis of performance of facial recognition based on AI for disease diagnosis.

### 2.4. Statistical Methods

Extracted two-by-two data are graphically shown in a forest plot with the point estimate of sensitivity and specificity and their 95% CIs. Considering the unclear and heterogeneous thresholds for diagnosing different disease with facial phenotypes by facial recognition methods, we used a quantitative random-effects model with bivariate mixed-effects binary regression to combine the sensitivity and specificity and to estimate the summary receiver operating characteristic (SROC) curve. The combined SROC curve and the optimum diagnostic threshold with 95% confidence region and 95% prediction region were plotted. Subgroup analyses and meta-regression were used to explore the heterogeneity between studies. Facial recognition intensity (FRI) and sample size of the training set were analyzed as covariates in meta-regression to explore quantitative relationships with diagnostic accuracy of facial recognition. The result of the meta-regression is shown in a bubble chart and demonstrates a fitting straight line. In addition to FRI and sample size of the training set, we also estimated the following covariates in subgroup analysis: resource of the control group, photo resolution, number of included diseases, and model of facial recognition. Covariates with statistically significant coefficients were regarded as a source of heterogeneity. The robustness of the main results was evaluated by sensitivity analyses. We explored the effect of excluding studies not reporting the model of facial recognition or gold standard of targeted conditions and those using internal validation to evaluate the models.

Data analysis for this paper was performed using Stata Statistical Software 16 (StataCorp., College Station, TX, USA) with two-tailed probability of Type I error of 0.05 (α = 0.05).

## 3. Results

### 3.1. Systematic Review

Figure 1 shows the flow diagram for filtering articles. We identified 2534 records by electronic search and 29 by hand search. In total, 141 full-text articles were assessed for eligibility, and 20 studies in 14 publications met our criteria for inclusion. Ozdemir et al. [19] included three studies, and Basel-Vanagaite et al. [20], Gurovich et al. [2], Zhao et al. [17], and Saraydemir et al. [16] included two studies using different sample sets in one publication.

The detailed characteristics of the eligible studies are shown in Appendix A. The total number of subjects tested in the included studies was 12,557. A single disease was targeted in 16 studies, including 3 studies on Cornelia de Lange syndrome [2,20], 2 on Turner syndrome [21,22], 3 on Down syndrome [16,17], 1 on Angelman syndrome [2], 4 on acromegaly [23,24,25,26], 2 on Cushing’s syndrome [27,28], and 1 study on fetal alcohol spectrum disorders (FASD) [29], as multiple diseases were detected in 4 studies [17,19]. Nine studies used photographs from public databases and web pages [2,25,27], and 11 studies obtained their photographs in local hospitals [20,21,22,23,24]. Ten studies described the demographic characteristics of their study population, reporting a percentage of males ranging from 0 to 66.2% [16,17,21,22,24,25,26]. The diagnostic criteria of the targeted diseases were reported in 12 studies and included analysis of gene mutation [2,20] and karyotype [16,17,21,22], success of previous treatment [23], experts’ opinions [26], diagnostic tests [24,27,29]. An internal validation set was used for evaluation of the model in 12 studies [16,17,19,21,26,27,28,29], and an external validation set was reported in 8 studies [2,20,22,23,24,25]. Nine studies included a healthy control group [2,17,19,20,22], and patients with other diseases were included in 11 studies as a control group [16,17,21,23,24,25,26,27,28,29]. Apart from 5 studies not reporting the used AI architecture [17,19,20,26,27], several types of machine learning models were applied in 15 studies, including 7 studies using algorithms of deep learning and neural network [2,20,22,28,29] or a combination of neural network and other models [24]. The following models were also reported: SVM [16,21,23], Haar cascade classifier [25], hierarchical decision tree [19], k-NN [16,19] and combination of conventional models [11]. Fourteen studies reported a resolution of photographs ranging from 100 × 100 to 1500 × 1000 pixels [2,16,17,19,21,22,24,25,26,28]. The number of photographs used to train the model was reported in 20 studies and ranged from 30 to 3465, whereas the number of photographs in the testing set ranged from 17 to 242 [2,16,17,19,20,21,22,23,24,25,26,27,28,29].

### 3.2. Risk of Bias Assessment of the Eligible Studies

Appendix A show the results of the risk of bias assessment of the included studies. Regarding patient selection, risk of bias was unclear in 4 studies due to the insufficient information describing the sampling method [2,20] and high in 16 studies with a case–control design [16,17,19,21,22,23,24,25,26,27,28,29]. With respect to the index test, facial recognition was based on artificial intelligence algorithms without knowledge of the clinical diagnosis in all studies. As for the reference standard, risk of bias was low in 15 studies [2,16,17,20,21,22,24,26,27,28,29] and unclear in 5 studies that did not report the reference standard or an interpretation [19,23,25]. In the domain of flow and timing, risk of bias was low in 16 studies [2,16,17,20,21,22,23,25,26,27,28,29], unclear in 3 studies that did not report the reception of the reference standard [19], and high in 1 study because not all patients were subjected to the two tests assessed in the study [24].

### 3.3. Meta-Analysis

Figure 2 shows the paired forest plot for sensitivity and specificity with the corresponding 95% CIs for each study. Eligible studies were further combined, and the summary receiver operating characteristic (SROC) curve is shown in Figure 3 with the 95% confidence region and 95% prediction region. We calculated the following summarized estimates using random-effects models with 95% confidence interval (CI): sensitivity 89% (95% CI 82% to 93%), specificity 92% (95% CI 87% to 95%), positive likelihood ratio 11.1 (95% CI 6.5 to 18.8), negative likelihood ratio 0.12 (95% CI 0.08 to 0.20), and diagnostic odds ratio (OR) 90 (95% CI 35 to 230).

### 3.4. Sensitivity Analysis

After excluding eight studies that evaluated the models with an external validation set [2,20,22,23,24,25], pooled sensitivity was 86% (95% CI 75% to 93%), and specificity was 90% (95% CI 82% to 95%). After excluding studies with unclear models [17,19,20,26,27], pooled sensitivity was 90% (95% CI 83% to 94%), and specificity was 91% (95% CI 84% to 96%). After excluding studies with an unclear reference standard [17,20,25,28], pooled sensitivity was 89.0% (95% CI 82.0% to 94.0%), and specificity was 93.0% (95% CI 88.0% to 96.0%). Since these estimates were similar to the main results for the whole dataset, we did not find evidence that the overall combined estimates were influenced by external validation sets, unclear models, or unclear reference standards.

### 3.5. Evaluation of Facial Recognition Intensity (FRI)

Table 1 shows the prevalence, facial phenotypes of disease, and maximum penetrance of the phenotypes in the eligible studies. Among 16 studies targeting a single disease, Down syndrome showed 9 specific facial phenotypes, and the maximum penetrance of the facial phenotypes was 100% [18]; hence, the calculated FRI of Down syndrome was 9. As for Cornelia de Lange syndrome [2,20], it showed nine facial phenotypes, and the maximum penetrance was 82.7% according to the international consensus statement [30]. After calculation, FRI of Cornelia de Lange syndrome was 7.443. Angelman syndrome showed six facial features, with maximum penetrance of facial phenotypes of 100% and FRI of 8. Turner syndrome showed six facial phenotypes and the maximum penetrance of facial phenotypes was 56% [31]; therefore, FRI of Turner syndrome was 3.36. Fetal alcohol spectrum disorders (FASD) were associated with four facial phenotypes with maximum penetrance of 100% [29], resulting in FRI of 4.

Among endocrine diseases, acromegaly showed eight facial phenotypes [28]. Since the maximum penetrance was 100%, FRI of acromegaly was 8. Cushing’s syndrome showed five facial phenotypes and maximum penetrance of facial phenotypes of 100% [27,28], resulting in FRI of 5.

### 3.6. Effect of FRI on the Accuracy of Facial Recognition

Table 2 shows the results of random-effects model meta-regression analysis exploring the relationship between facial recognition intensity (FRI), sample size of the training set, and diagnostic accuracy of facial recognition. The coefficient of FRI in the model was 0.4868 (95% CI 0.0935 to 0.8800, *p* = 0.015), revealing a significant association with natural logarithms of OR of automatic diagnosis by facial recognition. Meanwhile, the sample size of the training set was not associated with diagnostic accuracy of facial recognition, indicating no significant contribution to the heterogeneity between studies.

Therefore, after excluding the sample size of the training set from the model, the relationship between facial recognition intensity and diagnostic accuracy of facial recognition was determined as shown in Figure 4. The model with FRI as a variable showed significant association with natural logarithms of OR of automatic diagnosis, with the coefficient of FRI corresponding to 0.4960 (95% CI 0. 0748 to 0.9171, *p* = 0.021), indicating that a larger FRI value of a disease was significantly associated with a higher diagnostic accuracy by facial recognition. The relationship between FRI value for a disease and diagnostic accuracy is shown in Equation (2):ln (OR) = ln [Se Sp/((1 − Se) × (1 − Sp))] = 0.4960 × FRI + 1.459(2)

According to Equation (2), Table 3 shows the quantitative association between FRI and accuracy of automatic diagnosis by facial recognition. When both sensitivity and specificity reached 85%, it was required that the FRI value of a disease reached 4.05. When sensitivity and specificity rose to 90%, FRI should correspondingly increase to 5.92. FRI needed to reach 8.93 to ensure that the sensitivity and specificity reached 95%.

### 3.7. Effect of Sample Size of the Training Set and AI Model on the Accuracy of Facial Recognition

Table 4 lists the range of FRI, sample sizes of the training set, AI models, as well as relative median and range of diagnostic accuracy by facial recognition. As for the sample size of the training set, which ranged from 30 to 3465 in the eligible studies, it was shown that the diagnostic accuracy of diseases with FRI higher than 8 was greater than 0.95, even if the sample size of the training set was lower than 100, with the minimum sample size being 30. Diseases with FRI ranging from 6 to 8 showed relatively low diagnostic accuracy when the sample size of the training set was lower than 100, with the minimum sample size being 49, and the accuracy increased with the sample size. The minimum training size for diseases with FRI lower than 6 was 60, and a sample size greater than 1000 significantly improved the diagnostic accuracy of facial recognition, indicating that a modest increase in the sample size of the training set played an important role in improving the diagnostic accuracy of diseases with low FRI.

AI methods also showed a similar trend. Diagnostic accuracy of AI reached more than 0.95 with non-deep learning models for diseases with FRI higher than 8, and the application of deep learning models contributed to a higher sensitivity for diseases with lower FRI. Especially for diseases with FRI lower than 6, the median sensitivity improved from 0.688 to 0.929 by using deep learning models. However, the specificity was not influenced by the use of deep learning models.

### 3.8. Sources of Heterogeneity

Table 5 shows the detailed results of subgroup analyses exploring the potential source of between-study heterogeneity. Facial feature strength was significantly associated with diagnostic accuracy by facial recognition (*p* = 0.003). However, we found no association between facial recognition’s accuracy and photographic resolution, sample size of training sets, model of machine learning, number of targeted diseases, and selection of the control group.

## 4. Discussion

At present, artificial intelligence methods have been widely applied in different fields. However, studies exploring factors influencing the diagnostic accuracy of these methods, as well as systematic reviews and meta-analyses summarizing AI application in the diagnosis of heterogeneous diseases are still lacking. To our knowledge, this is the first study that fills this gap by summarizing heterogeneous studies on the automatic diagnosis of diseases on the basis of facial features and quantitatively analyzes the diagnostic capability of facial recognition based on AI. The review and meta-analysis were conducted strictly following the guidelines for diagnostic reviews [32]. Comprehensive and large-scale studies published so far were included, searched in both medical databases and engineering and technology databases. Representative and high-quality studies focused on different diseases using various known AI methods and were conducted in different countries. Our study summarized and quantitatively analyzed heterogeneous studies on the automatic diagnosis of different diseases based on facial features, showing a pooled sensitivity of 89% (95% CI 82% to 93%) and a specificity of 92% (95% CI 87% to 95%), similar to the results of previous meta-analyses on automatic image recognition for diabetic retinopathy screening [8,33,34], colorectal neoplasia, and breast cancer [35,36,37,38], indicating a promising diagnostic performance of facial recognition based on AI for heterogeneous diseases. A sensitivity analysis was conducted to evaluate the robustness of the results. The results were interpreted logically and adapted to clinical applications.

We propose a new index, facial feature intensity (FRI), to reflect the complexity of facial features associated with a targeted object. FRI was defined to minimize the heterogeneity across objects in AI applications and is calculated by multiplying the number of independent facial phenotypes by the maximum penetrance of these facial phenotypes. The number of details in facial features determines the complexity that distinguishes facial features of the targeted object from those of other objects, and the penetrance is the proportion of patients showing a certain complexity of facial features. Since FRI was revealed as the most important influencing factor for the diagnostic accuracy of facial recognition based on AI, the complexity of a targeted object plays the most important role in AI performance, rather than AI technology itself. According to Equation (2) in the meta-regression analysis, the expected accuracy of facial recognition for detecting a disease with the known FRI value could be predicted by calculation, which is of great clinical value.

The interactions between AI parameters and FRI were also taken into consideration, including sample size of the training set and AI architecture. The results revealed that, although larger training size and selection of deep-learning models did not contribute significantly to the heterogeneity between studies in either meta-regression or subgroup analysis, they showed a trend indicating improved diagnostic accuracy for diseases with lower FRI. An appropriate increase in the size of the training samples and the use of deep-learning models improved the accuracy of facial recognition, revealing that the improvement of AI parameters contributed to a better performance of AI for objects with low complexity. This finding is also supported by results on the detection of breast cancer, showing that increasing the training set size would not increase the diagnostic accuracy continuously [38]. Since the number of patients with rare diseases is limited, this finding is clinically significant as it indicates that the sample size of the training set can be within reasonable limits in AI applications. Moreover, the existing AI models have still to be improved to increase the diagnostic accuracy by facial recognition. Therefore, technology innovation is needed, and new AI methods might show better diagnostic accuracy by facial recognition.

Moreover, according to our findings, we propose a new hypothesis regarding AI application, that we named object’s complexity theory (OCT) and that could be expanded to the application of AI technology in other fields. According to OCT, within the limits of a reasonable research design, the complexity of the targeted objects determines the complexity of AI processing and plays the most important role in AI performance, while improvement of AI parameters contributes to a better performance of AI for objects with low complexity. The hypothesis is consistent with existing evidence and is supported by previous theorems. According to the complexity theory proposed by J. Hartmanis and R. E. Stearns in 1965, the deep commonalities typical of complex systems determine the process of solving problems, which is relevant in diverse fields [39]. OCT represents the development and extension of the complexity theory regarding the performance of AI applications. According to the No Free Lunch Theorem (NFLT) for artificial intelligence proposed by David Wolpert and William Macready in 1996 [40] and optimized in 1997 [41], an algorithm performing well on a certain object paid with degraded performance on all remaining objects. If we use *i* to index the examined objects arbitrarily and O*_i_* to represent an object, the NFLT is represented by Equation (3)
(3)$$\underset{k}{\sum }f\left(O_{k},a_{i}\right)=\underset{k}{\sum }f\left(O_{k},a_{j}\right),\;\forall i,j$$
where *a_i_* and *a_j_* are algorithms, and $f\left(O_{k},a_{i}\right)$ is the performance of *a_i_* on the object *O_k_*. The equation shows that the overall performances of all the algorithms were the same. The only way a strategy could outperform another is to specialize the structure of the specific object under consideration [42]. As for our hypothesis, OCT, based on the application in facial recognition, we can establish Equation (4), on the basis of NFLT:(4)$$f\left(O_{k},a_{i}\right)=g\left(O_{k},{\mathrm{FRI}}_{k}\right),\;\forall i,{\;\mathrm{if}\;\mathrm{FRI}}_{k}\geq 6$$
where $g\left(O_{k},{\mathrm{FRI}}_{k}\right)$ is the performance of the algorithm *a_i_* on the *k*-th object. The equation revealed that the structure of the object is reflected in the FRI. For objects with a large enough FRI, independently of the parameters of AI technology, the performances are more or less the same. The theory provides a new idea, suggesting that more indices for the evaluation of the complexity of targeted objects should be explored and developed in further studies to better determine AI performance in other fields.

Moreover, OCT and its application in facial recognition provide a new idea to deal with heterogeneity in studies and to evaluate the complexity of targeted objects. OCT should be applied and developed in further studies to determine AI performance in other fields. For image recognition based on AI, facial feature intensity (FRI) could also be converted into image feature intensity (IRI) to describe the characteristics of images related to more diseases. IRI might be the most important factor for AI performance within the limits of a reasonable sample size and of the study design. Previous studies have demonstrated that the image characteristics of diseases play an important role in the performance of image recognition by AI methods [43], including the automatic screening of pulmonary nodules [7,44,45], referable glaucomatous optic neuropathy (GON) [46], colorectal adenoma and polyps [47,48], which also indicates that IRI describes image characteristics of diseases and is critical for AI performance in automatic image recognition. As has been shown before for diabetic retinopathy screening, no statistically significant contribution to heterogeneous diagnostic accuracy has been demonstrated for sample size of the training sets and architecture of convolutional neural networks [34]. Therefore, the complexity theory explains the relationship between complexity of a disease and AI performance and should be extended to other AI applications.

There are some limitations in our study. First, the photographs overlapped in several studies using the same data sources, and it was difficult to eliminate this and evaluate its influence. Second, the risk of bias for the domain of patient selection was high or unclear in several studies. More than half of the studies had a case–control design, due to the limited number of patients with rare diseases. In addition, no traditional thresholds were mentioned in these studies, and we could only compare the sensitivity and specificity by finding the best cut-off point.

## 5. Conclusions

We quantitatively analyzed studies on the association of heterogeneous diseases with facial features and revealed the promising diagnostic performance of facial recognition based on AI in detecting diseases on the basis of facial features. A new index, facial feature intensity (FRI), was proposed to describe the complexity with facial features associated with different diseases, which was proved to be the most important factor influencing diagnostic accuracy by facial recognition. In addition, we explored the universal rules governing facial recognition based on AI in the field of medical diagnosis and provide a potential reference to solve practical problems in AI applications. An appropriate increase in training sample size and the use of deep learning models might play a role in improving the diagnostic accuracy for diseases with lower FRI. Our study firstly proposes a new hypothesis, the object’s complexity theory (OCT), on the performance of AI and provides a new idea for dealing with heterogeneity when evaluating AI performance in other applications.

## Figures and Tables

**Figure 1 jpm-11-01172-f001:**
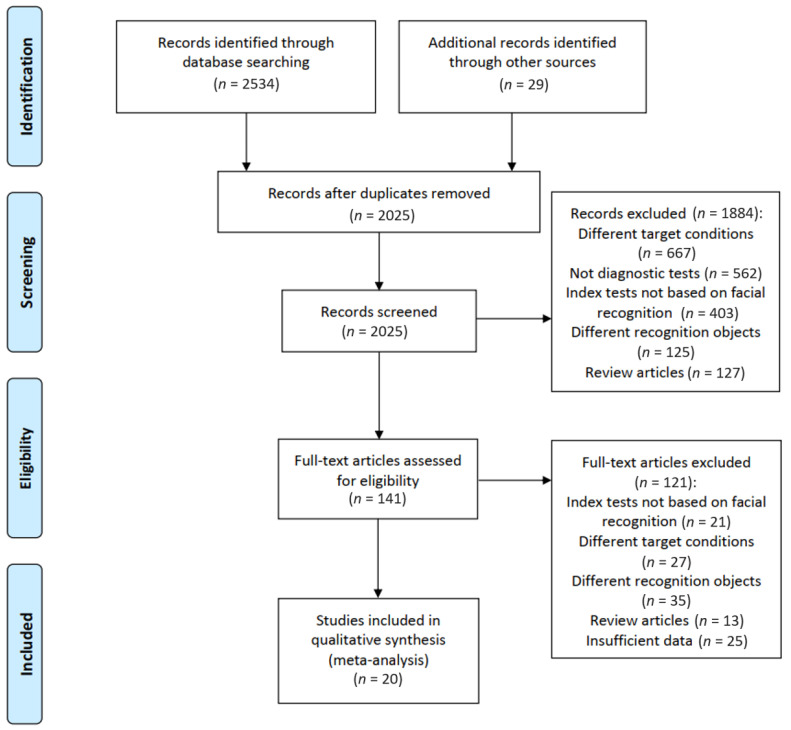
Flow chart for study inclusion and exclusion. The titles and the abstracts were screened by two reviewers independently, and the full texts of potentially eligible studies were further screened.

**Figure 2 jpm-11-01172-f002:**
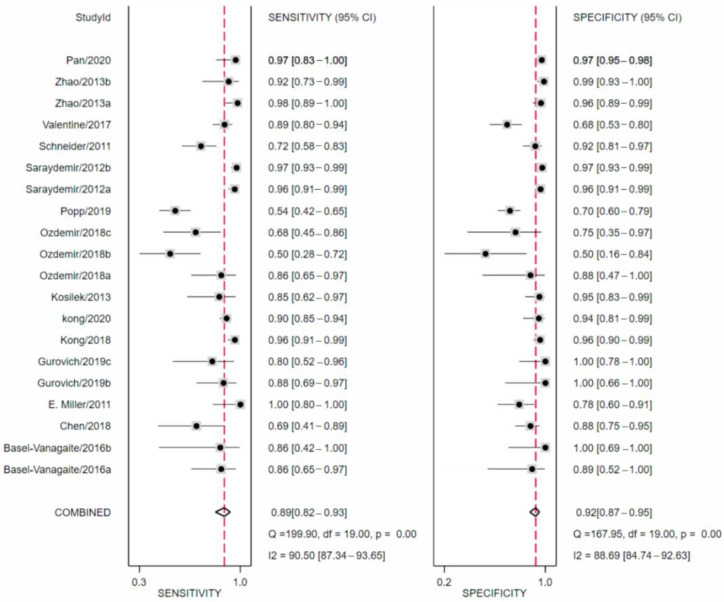
Forest plots of sensitivity and specificity in automatic diagnosis by facial recognition.

**Figure 3 jpm-11-01172-f003:**
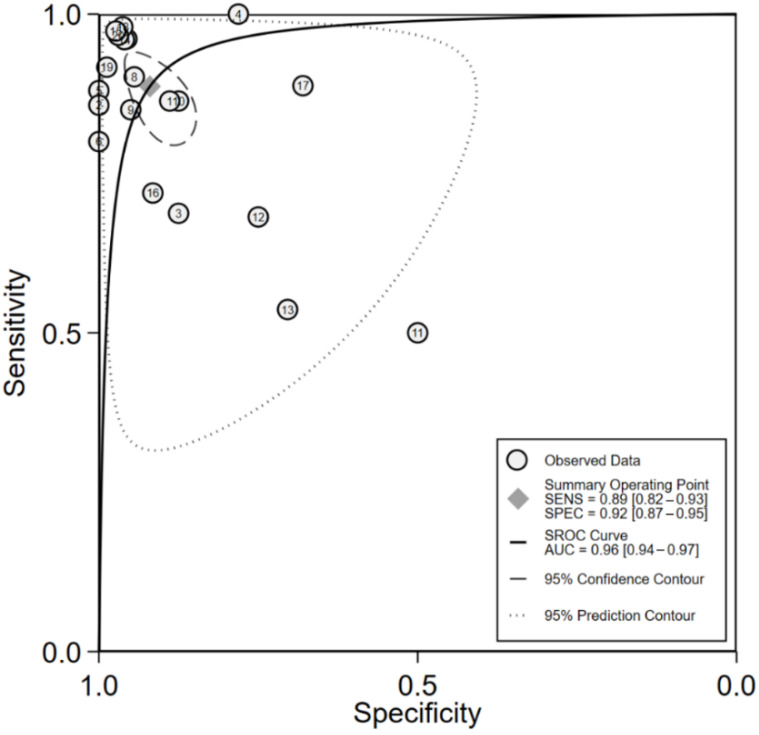
Summary receiver operating characteristics (SROC) curves of eligible studies. The dashed line indicates the 95% confidence region, and the dotted line indicates the 95% prediction region.

**Figure 4 jpm-11-01172-f004:**
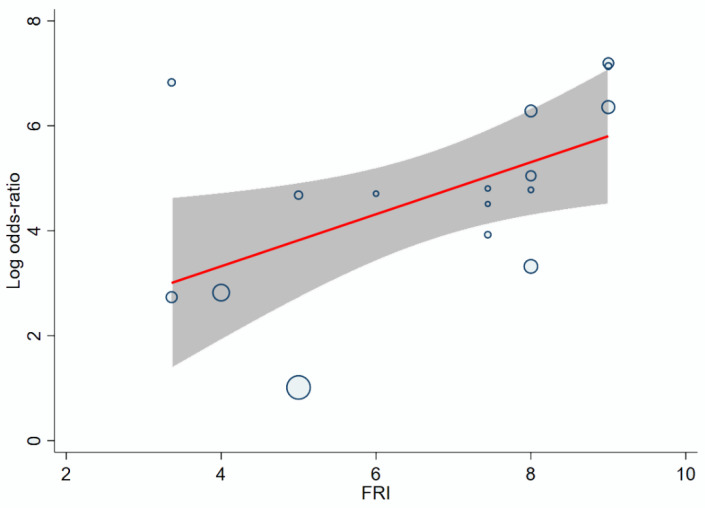
Bubble plots of meta-regression between FRI and ln(OR) of automatic diagnosis by facial recognition. FRI = facial recognition intensity, OR = diagnostic odds ratio. The straight line indicates linear prediction in the meta-regression model between FRI and diagnostic accuracy. The gray zone indicates the 95% confidence region, and the round bubbles represent the eligible studies. The size of the bubbles indicates the impact on the model.

**Table 1 jpm-11-01172-t001:** Assessment of facial recognition intensity (FRI) of diseases in the eligible studies.

Disease	Prevalence	Maximum Penetrance (Pmax)	Facial Phenotypes		Facial Recognition Intensity (FRI)
			Independent Facial Phenotypes	Number of Facial Phenotypes (Nf)	
Down syndrome [16,17]	1/300~1000	100%	Short face Upward slanting eyes Epicanthus Brushfield spots (white spots on the colored part of the eyes) Low-set ears Small ears Flattened nose Small mouth Protruding tongue	9	9
Acromegaly [23,24,25,26]	7/1000	100%	Forehead bulge Prominent jaw Prominent zygomatic arch Deep nasolabial folds Enlarged nose Enlarged brow Enlarged ear Enlarged lip	8	8
Cornelia de Lange Syndrome [2,20]	1/10,000~1/30,000	82.7%	Short face Small jaw Arched eyebrows Joined eyebrows Short nose Forward nostril Long philtrum Thin upper lip Upturned corners of the mouth	9	7.443
Angelman syndrome [2]	1/20,000~1/12,000	100%	Narrow bifrontal diameter Huge jaw Almond-shaped palpebral fissures Narrow nasal bridge Thin upper lip Protruding tongue	6	6
Cushing’s syndrome [27,28]	4/100,000	100%	Red face Full moon face Acne Excessive hair Chemosis conjunctiva	5	5
Fetal alcohol spectrum disorders (FASDs) [29]	7.7/1000	100%	Small head Short palpebral fissures Smooth philtrum Thin vermilion border of the upper lip	4	4
Turner syndrome [21,22]	1/2500	56%	Small jaw Epicanthus Ptosis Ocular hypertelorism Low-set ears Multiple facial nevi	6	3.36

**Table 2 jpm-11-01172-t002:** Meta-regression between FRI, sample size of the training set, and ln(OR) of automatic diagnosis by facial recognition. FRI = facial recognition intensity, OR = diagnostic odds ratio. FRI and sample size of the training set were analyzed as covariates in a meta-regression model to explore the heterogeneity between studies. Their coefficient and 95% confidence interval in the model are shown with two-tailed probability of type I error of 0.05 (α = 0.05).

Covariate	Coefficient [95 Cl]	*p* Value
Facial recognition intensity (FRI)	0.4939 [0.0710,0.9169]	0.022
Sample size of the training set	0.0004 [−0.0006,0.0014]	0.467

**Table 3 jpm-11-01172-t003:** Association between FRI and accuracy of automatic diagnosis by facial recognition. FRI = facial recognition intensity, OR = diagnostic odds ratio. Quantitative relationship between FRI and diagnostic accuracy (including Figure 2. in meta-analysis. ln (OR) = ln [Se Sp/(1 − Se) (1 − Sp)] = 0.4951 × FRI + 1.46.

Sensitivity	Specificity	OR	ln(OR)	FRI
85%	85%	32.11	3.47	4.05
90%	85%	51.00	3.93	4.98
90%	90%	81.00	4.39	5.92
95%	90%	171.00	5.14	7.42
95%	95%	361.00	5.89	8.93

**Table 4 jpm-11-01172-t004:** Association between FRI, sample size of the training set, AI models, and accuracy of automatic diagnosis by facial recognition. FRI = facial recognition intensity, DL = deep learning. The diagnostic accuracy is shown as median (minimum, maximum).

FRI	Minimum Sample Size of Training Set	Range of Sample Size of Training Set	Range of Accuracies		AI Models	Range of Accuracies	
			Sensitivities	Specificities		Sensitivities	Specificities
>8	30	<100	0.967 (0.960~0.973)	0.967 (0.960~0.973)	Non-DL	0.973 (0.960~0.977)	0.962 (0.960~0.973)
		100~200	0.977	0.962			
6~8	49	<100	0.710	1.000	Non-DL DL	0.810 (0.719~0.901) 0.860 (0.800~0.960)	0.972 (0.944~1.000) 1.000 (0.890~1.000)
		100~1000	0.790 (0.719~0.860)	0.903 (0.890~0.915)			
		>1000	0.901 (0.800~0.960)	1.000 (0.944~1.000)			
<6	60	<100	0.769 (0.688~0.850)	0.913 (0.875~0.950)	Non-DL	0.688	0.875
		100~1000	0.714 (0.537~0.890)	0.697 (0.690~0.704)	DL	0.929 (0.890~0.967)	0.830 (0.690~0.970)
		>1000	0.967	0.970			

**Table 5 jpm-11-01172-t005:** Subgroup analyses for the accuracy of automatic diagnosis by facial recognition. Image resolution was calculated by multiplying column pixels by row pixels. If images of different resolution were used, the average resolution was calculated. The two-tailed probability of type I error was 0.05 (α = 0.05).

Subgroup Variables	Numbers of Eligible Studies	Sensitivity, % [95 Cl]	Specificity, % [95 Cl]	*p* for Interaction
Image resolution				0.415
<30,000 pixels	7	0.85 [0.73–0.97]	0.90 [0.82–0.98]	
≥30,000 pixels	7	0.90 [0.82–0.98]	0.94 [0.89–0.98]	
Sample size of training set				0.145
<1000	14	0.87 [0.80–0.93]	0.89 [0.84–0.95]	
≥1000	6	0.92 [0.86–0.99]	0.97 [0.93–1.00]	
Model/system of AI				0.802
Neural network	7	0.91 [0.83–0.99]	0.93 [0.85–1.00]	
Non-neural network	8	0.92 [0.86–0.97]	0.92 [0.86–0.98]	
Number of diseases				0.930
1	16	0.90 [0.86–0.95]	0.78 [0.60–0.97]	
>1	4	0.93 [0.89–0.97]	0.88 [0.74–1.00]	
Selection of control group				0.573
Healthy	9	0.85 [0.75–0.95]	0.94 [0.89–0.99]	
Other diseases	11	0.90 [0.84–0.96]	0.91 [0.86–0.97]	
Facial recognition intensity (FRI)				0.003
≤6	7	0.81 [0.71–0.90]	0.90 [0.83–0.96]	
>6	9	0.95 [0.92–0.98]	0.95 [0.91–0.98]	

## Data Availability

Data from this study will be made available upon request from the authors.

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
