# Peer review of "Facial Recognition Intensity in Disease Diagnosis Using Automatic Facial Recognition"

_jpm, 2021, doi:10.3390/jpm11111172_

Round 1

Reviewer 1 Report

The authors conducted a systematic review and analysis to summarize the studies in heterogeneous diseases using facial features and to explore the universal rules in facial recognition based on artificial intelligence in the field of medical diagnosis. Studies in heterogeneous diseases with facial features was summarized and quantitatively analyzed. Moreover, a new index, facial recognition intensity (FRI), was proposed to describe the complexity of diseases with facial phenotypes. The diagnostic accuracy was analyzed quantitatively.

The manuscript’s topic is interesting and the paper presents some new findings. The manuscript is easy to follow. However, some major issues need to be addressed before accepting it for publication.

  1. Your title is too long and should be shortened. A good title should not exceed 10 words (up to 12 if necessary).
  2. The title should be clear and informative, and should reflect the aim and approach of the work.
  3. What is the abbreviation QUADAS-2 in page 3 stands for?
  4. Extensive English revision grammatical checks need to be conducted. For example, was showed …should be was shown. This was repeated many times.

- additionally, sentences in lines 147-152 need to be rephrased and corrected.

  1. Equation in line 374 needs to be numbered and corrected, as well as all the used symbols need to be written in a correct way.
  2. Equation in line 381 needs to be numbered and corrected, as well as all the used symbols need to be written in a correct way.
  3. In lines 324-325, the authors mentioned the obtained results in previous meta-analyses on automatic image recognition for diabetic retinopathy screening as in [27,28], please refer to the up-to-date results for diabetic retinopathy detection presented in this study: Diabetic Retinopathy Diagnosis from Fundus Images Using Stacked Generalization of Deep Models [IEEE Access, 2021], which obtained very good results in the subject of this paper.
  4. The authors should consider to insert the following diagnostic models using deep learning approaches, which proved to be very successful diagnostic systems for automatic screening of pulmonary disease as COVID-19: COVID-Nets: deep CNN architectures for detecting COVID-19 using chest CT scans [PeerJ Computer Science, 2021]; Explainable covid-19 detection using chest ct scans and deep learning [Sensors, 2021], which also provides useful insights on how to obtain explainable diagnostic systems.
  5. References need to be updated with recent works and to be in a unified format. Also complete missing information as in Ref. 26.

Author Response

Comment 1: Your title is too long and should be shortened. A good title should not exceed 10 words (up to 12 if necessary).

Response: Thank you for your kind reminding. According to your instruction, we have shortened the title to less than 10 words. 1.   The previous title “The influence of a new index, facial recognition intensity, on diagnostic performance of automatic facial recognition based on artificial intelligence and proposal of object’s complexity theory” has been shortened into“Facial recognition intensity in diagnosis using automatic facial recognition”.

Comment 2: The title should be clear and informative, and should reflect the aim and approach of the work.

Response: The title has been revised into “Facial recognition intensity in diagnosis using automatic facial recognition”, which reflect facial recognition intensity (FRI), the novel index to reduce heterogeneity, as the most important innovation in approach of our work.

Comment 3: What is the abbreviation QUADAS-2 in page 3 stands for?

Response: We were really sorry for carelessly not explaining the abbreviation in our manuscript. QUADAS-2 stands for Quality Assessment of Diagnostic Accuracy Studies-2, which is a revised tool for the quality assessment of diagnostic accuracy studies[1]. It has been widely used in systematic reviews and meta-analyses of diagnostic accuracy. The QUADAS-2 tool is applied in 4 phases: summarize the review question, tailor the tool and produce review-specific guidance, construct a flow diagram for the primary study, and judge bias and applicability. And we have added the full name in manuscript line 104.

Comment 4: Extensive English revision grammatical checks need to be conducted. For example, was showed …should be was shown. This was repeated many times.

- additionally, sentences in lines 147-152 need to be rephrased and corrected.

Response: Thank you for pointing this weakness out. According to your suggestion, we have revised the whole manuscript carefully to avoid any grammar or syntax error and to make sure sentences fluent and clear. In addition, we have consulted native English speakers to check the written English before the submission this time. A lot of corrections have been made. Part of the corrections has been listed below and the detailed changes are marked in the revised manuscript.

  1. The previous sentence “Studies in heterogeneous diseases with facial features was firstly summarized and quantitatively analyzed. ” has been revised into “Studies in heterogeneous diseases with facial features were firstly summarized and quantitatively analyzed. ”(line 30-31)
  2. The previous sentence “The calculation of FRI was showed in equation (1), by multiplying the number of independent facial phenotypes of disease and the maximum penetrance among these facial features.” has been revised into “The calculation of FRI was shown in equation (1), by multiplying the number of independent facial phenotypes of disease and the maximum penetrance among these facial features.” (line 111-113)
  3. The previous sentence “The detailed characteristics of eligible studies were showed in Supplementary Table 2.” has been revised into “The detailed characteristics of eligible studies were shown in Supplementary Table 2.” (line 164-165)
  4. The previous sentence “Eligible studies were further combined and the summary receiver operating characteristic (SROC) curve is shown in Figure 3 with the 95% confidence region and 95% prediction region.” has been revised into “Eligible studies were further combined and the summary receiver operating characteristic (SROC) curve was shown in Figure 3 with the 95% confidence region and 95% prediction region.” (line 203-205)

In addition, the previous sentences in lines 147-152 has been rephrased and corrected according to your good advice. The previous sentences “Figure 1 showed the flow diagram for filtering articles. We identified 2534 records by electronic search and 29 by hand search. 141 full-text articles were assessed for eligibility.20 studies in 14 publications met our criteria for inclusion. Ozdemir et al. included three studies and Basel-Vanagaite et al., Gurovich et al., Zhao et al., and Saraydemir et al. included two studies using different sample sets in one publication, respectively” have been changed into “Figure 1 showed the flow diagram for filtering articles. We identified 2534 records by electronic search and 29 by hand search. 141 full-text articles were assessed for eli-gibility.20 studies in 14 publications met our criteria for inclusion. Ozdemir et al. included three studies. Basel-Vanagaite et al., Gurovich et al., Zhao et al., and Saraydemir et al. included two studies using different sample sets in one publication, respectively.”(line 153-158)

Comment 5: Equation in line 374 needs to be numbered and corrected, as well as all the used symbols need to be written in a correct way.

Response: Thank you for your reminding. We were really sorry for our careless mistakes. The previous equation in line 374 was the explanation of the no free lunch theorem (NFLT). It has been numbered and corrected as followed:

“If we use i to index the objects arbitrarily we are concerned, and Oi to represent the object, the NFLT said as equation (3), (Please see the attachment.)

where ai and aj were algorithms, and  was the performance of ai on the object Ok.”(line 379-384)

Comment 6: Equation in line 381 needs to be numbered and corrected, as well as all the used symbols need to be written in a correct way.

Response: The previous equation in line 374 has been numbered and corrected as followed:

“As for our hypothesis, OCT, based on the application in facial recognition, it was plausible that we had a stricter version as equation (4) extended from the NFLT, that was, (Please see the attachment.)

where  meant the performance of the algorithm ai­ on the k-th object.” (line 386-392)

Comment 7: In lines 324-325, the authors mentioned the obtained results in previous meta-analyses on automatic image recognition for diabetic retinopathy screening as in [27,28], please refer to the up-to-date results for diabetic retinopathy detection presented in this study: Diabetic Retinopathy Diagnosis from Fundus Images Using Stacked Generalization of Deep Models [IEEE Access, 2021], which obtained very good results in the subject of this paper.

Response: Thank you for your advice and we have added this up-to-date article for diabetic retinopathy detection to our introduction and discussion. (seen in line 53-56 and 334-335)

Comment 8: The authors should consider to insert the following diagnostic models using deep learning approaches, which proved to be very successful diagnostic systems for automatic screening of pulmonary disease as COVID-19: COVID-Nets: deep CNN architectures for detecting COVID-19 using chest CT scans [PeerJ Computer Science, 2021]; Explainable covid-19 detection using chest ct scans and deep learning [Sensors, 2021], which also provides useful insights on how to obtain explainable diagnostic systems.

Response: We agree with your good advice that diagnostic systems for automatic screening of various diseases, especially in COVID-19. Both two articles you mentioned have been added to our introduction. (seen in line 53-56).

Comment 9: References need to be updated with recent works and to be in a unified format. Also complete missing information as in Ref. 26.

Response: Thank you very much for your constructive comments and suggestions. We have checked all the references carefully and have added several recent works as Ref 6-10&12. And we revised the references into unified MDPI format.

Ref. 26 has been confirmed and completed into “Higgins JPT, Thomas J, Chandler J, Cumpston M, Li T, Page MJ, Welch VA (editors). Cochrane Handbook for Systematic Reviews of Interventions. 2nd Edition. Chichester (UK): John Wiley & Sons, 2019.”(line 527-529)

Reviewer 2 Report

The research presented in the article is very interesting and opens great horizons in a world that must use technologies ethically as well as efficiently.
The implication of AI and deep learning can lead not only to diagnostic implications but also to the analysis of moods and how emotions can interact with the "physical" states of people.

Author Response

 Thank you very much for your time involved in reviewing the manuscript and your encouraging comments on the merits. We agree with you that AI implication can also lead to the analysis of moods and how emotions can interact with the "physical" states of people, with potential similar rules of AI performance. Our novel hypothesis, OCT, should be applied and developed in further studies for explanations of AI performance in other fields. 

Reviewer 3 Report

This paper is about a meta-analysis of facial recognition in disease diagnosis. Overall the paper is well written. 

The title of the paper is not straightforward to understand. It should be easier to understand. 

The method is finely described.

The conclusion is supported by the result. 

Some English usage can be improved. 

No need to define FRI multiple times through the paper. 

Equation on line 374 appears to be garbled, it should be double-checked by the authors.

Author Response

Comment 1: This paper is about a meta-analysis of facial recognition in disease diagnosis. Overall the paper is well written. The method is finely described. The conclusion is supported by the result.

The title of the paper is not straightforward to understand. It should be easier to understand.

Response: Thank you very much for your time involved in reviewing the manuscript and your encouraging comments on the merits. According to your good advice, we have shortened the title to make it clear and informative. The previous title “The influence of a new index, facial recognition intensity, on diagnostic performance of automatic facial recognition based on artificial intelligence and proposal of object’s complexity theory” has been revised into “Facial recognition intensity in diagnosis using automatic facial recognition”, which reflect facial recognition intensity (FRI), the novel index to reduce heterogeneity, as the most important innovation in approach of our work.

Comment 2: Some English usage can be improved.

Response: Thank you for pointing this weakness out. According to your suggestion, we have revised the whole manuscript carefully to avoid any grammar or syntax error and to make sure sentences fluent and clear. In addition, we have consulted native English speakers to check the written English before the submission this time. A lot of corrections have been made. Part of the corrections has been listed below and the detailed changes are marked in the revised manuscript.

  1. The previous sentence “Studies in heterogeneous diseases with facial features was firstly summarized and quantitatively analyzed. ” has been revised into “Studies in heterogeneous diseases with facial features were firstly summarized and quantitatively analyzed. ”(line 30-31)
  2. The previous sentence “The calculation of FRI was showed in equation (1), by multiplying the number of independent facial phenotypes of disease and the maximum penetrance among these facial features.” has been revised into “The calculation of FRI was shown in equation (1), by multiplying the number of independent facial phenotypes of disease and the maximum penetrance among these facial features.” (line 111-113)
  3. The previous sentences “Figure 1 showed the flow diagram for filtering articles. We identified 2534 records by electronic search and 29 by hand search. 141 full-text articles were assessed for eligibility.20 studies in 14 publications met our criteria for inclusion. Ozdemir et al.[13] included three studies and Basel-Vanagaite et al.[14], Gurovich et al.[2], Zhao et al.[11], and Saraydemir et al.[10] included two studies using different sample sets in one publication, respectively” have been changed into “Figure 1 showed the flow diagram for filtering articles. We identified 2534 records by electronic search and 29 by hand search. 141 full-text articles were assessed for eli-gibility.20 studies in 14 publications met our criteria for inclusion. Ozdemir et al.[13] included three studies. Basel-Vanagaite et al.[14], Gurovich et al.[2], Zhao et al.[11], and Saraydemir et al.[10] included two studies using different sample sets in one publication, respectively.”(line 153-158)
  4. The previous sentence “The detailed characteristics of eligible studies were showed in Supplementary Table 2.” has been revised into “The detailed characteristics of eligible studies were shown in Supplementary Table 2.” (line 164-165)
  5. The previous sentences “Figure 1 showed the flow diagram for filtering articles. We identified 2534 records by electronic search and 29 by hand search. 141 full-text articles were assessed for eligibility.20 studies in 14 publications met our criteria for inclusion. Ozdemir et al. included three studies and Basel-Vanagaite et al., Gurovich et al., Zhao et al., and Saraydemir et al. included two studies using different sample sets in one publication, respectively” have been changed into “Figure 1 showed the flow diagram for filtering articles. We identified 2534 records by electronic search and 29 by hand search. 141 full-text articles were assessed for eli-gibility.20 studies in 14 publications met our criteria for inclusion. Ozdemir et al. included three studies. Basel-Vanagaite et al., Gurovich et al., Zhao et al., and Saraydemir et al. included two studies using different sample sets in one publication, respectively.”(line 153-158)

Comment 3: No need to define FRI multiple times through the paper.

Response: Thank you for your advice and we have deleted the definition of FRI in line 242-245 to avoid duplication and maintain it twice in our method and discussion.

Comment 4: Equation on line 374 appears to be garbled, it should be double-checked by the authors.

Response: Thank you for your reminding. We were really sorry for our careless mistakes. The previous equation in line 374 was the explanation of the no free lunch theorem (NFLT). It has been numbered and corrected as followed:

“If we use i to index the objects arbitrarily we are concerned, and Oi to represent the object, the NFLT said as equation (3), (Please see the attachment.)

where ai and aj were algorithms, and  was the performance of ai on the object Ok.”(line 379-384)

Also, the previous equation in line 374 has been numbered and corrected as followed:

“As for our hypothesis, OCT, based on the application in facial recognition, it was plausible that we had a stricter version as equation (4) extended from the NFLT, that was, (Please see the attachment.)

where  meant the performance of the algorithm ai­ on the k-th object.” (line 386-392)

Round 2

Reviewer 1 Report

The authors have addressed all my comments and improved the manuscript significantly.